# Ocular Biodistribution Studies Using Molecular Imaging

**DOI:** 10.3390/pharmaceutics11050237

**Published:** 2019-05-16

**Authors:** Ana Castro-Balado, Cristina Mondelo-García, Miguel González-Barcia, Irene Zarra-Ferro, Francisco J Otero-Espinar, Álvaro Ruibal-Morell, Pablo Aguiar, Anxo Fernández-Ferreiro

**Affiliations:** 1Pharmacy Department, University Hospital of Santiago de Compostela (SERGAS), 15706 Santiago de Compostela, Spain; ana.castro.balado@gmail.com (A.C.-B.); crismondelo1@gmail.com (C.M.-G.); miguel.gonzalez.barcia@sergas.es (M.G.-B.); irene.zarra.ferro@sergas.es (I.Z.-F.); 2Pharmacology Group, Health Research Institute Santiago Compostela (IDIS), 15706 Santiago de Compostela, Spain; 3Department of Pharmacology, Pharmacy and Pharmaceutical Technology and Industrial Pharmacy Institute, Faculty of Pharmacy, University of Santiago de Compostela (USC), 15782 Santiago de Compostela, Spain; francisco.otero@usc.es; 4Nuclear Medicine Department, University Hospital of Santiago de Compostela (SERGAS), University of Santiago de Compostela, 15706 Santiago de Compostela, Spain; alvaro.ruibal.morell@sergas.es; 5Molecular Imaging Group, Health Research Institute Santiago Compostela (IDIS), 15706 Santiago de Compostela, Spain

**Keywords:** ocular biodistribution, ocular permanence, molecular imaging, PET, SPECT, MRI

## Abstract

Classical methodologies used in ocular pharmacokinetics studies have difficulties to obtain information about topical and intraocular distribution and clearance of drugs and formulations. This is associated with multiple factors related to ophthalmic physiology, as well as the complexity and invasiveness intrinsic to the sampling. Molecular imaging is a new diagnostic discipline for in vivo imaging, which is emerging and spreading rapidly. Recent developments in molecular imaging techniques, such as positron emission tomography (PET), single-photon emission computed tomography (SPECT) and magnetic resonance imaging (MRI), allow obtaining reliable pharmacokinetic data, which can be translated into improving the permanence of the ophthalmic drugs in its action site, leading to dosage optimisation. They can be used to study either topical or intraocular administration. With these techniques it is possible to obtain real-time visualisation, localisation, characterisation and quantification of the compounds after their administration, all in a reliable, safe and non-invasive way. None of these novel techniques presents simultaneously high sensitivity and specificity, but it is possible to study biological procedures with the information provided when the techniques are combined. With the results obtained, it is possible to assume that molecular imaging techniques are postulated as a resource with great potential for the research and development of new drugs and ophthalmic delivery systems.

## 1. Introduction

The access of drugs to the ocular structures through the route of systemic administration is greatly hampered by the anatomical barriers that separate the eyeball from the general circulation. This makes it necessary to systemically administer high doses of drugs in order to reach therapeutic concentrations in the internal structures of the eye and the vitreous humour, which can cause toxicity and adverse effects [1]. Accordingly, topical application is considered the most suitable route of administration for the patient. This is due to the fact that the ocular surface is an easily accessible area, especially regarding pathologies affecting the anterior segment [2,3]. However, conventional ophthalmic dosage forms present a low ocular bioavailability due to the multiple barriers of the eye, such as the cornea, the conjunctiva, the retinal pigment epithelium and the dynamic barriers (tear replacement and blood and lymphatic drainage mechanisms). To improve topical administration in the anterior segment, several devices and systems for sustained drug release have been studied, such as polymers, colloidal systems, emulsions, cyclodextrins, collagen and contact lenses [4,5,6,7,8]. Studying and identifying new ophthalmic administration systems helps to improve the application and permanence of drugs, offering dosage regimens that increase the adherence of patients [9].

Moreover, the intraocular administration of drugs is more complex and is reserved for the treatment of intraocular pathologies in which concentrations of drugs are not reached, either topically or systemically. In this case, the intravitreal, periocular, subretinal and suprachoroidal routes of administration are the most commonly used [10]. The diversity of factors affecting biodistribution, the invasiveness and the risks associated with the technique and sampling (vitreous haemorrhage, retinal detachment and endophthalmitis) limit clinical pharmacokinetic research, making it difficult to study appropriate dosages. Furthermore, preclinical ocular pharmacokinetic studies usually involve sacrificing the animals at different times after administration and analysing different sections of the eye, with the risk of redistributing the compound due to the dissection methods [11]. This is inconvenient and costly, and requires a large number of animals in each study.

To successfully develop new methods and techniques of ophthalmic drug delivery, it is important to have reliable ocular pharmacokinetic data. The complex and unique eye anatomy, as well as the invasiveness of the sampling, lead to the lack of complete knowledge of the ocular mechanisms for drug clearance, and therefore, for its pharmacokinetics [12,13]. In recent years, molecular imaging techniques have become a turning point for the development and characterisation of new drugs and formulations due to the fact that they are non-invasive and allow determining the distribution of the studied compound in real time.

## 2. Molecular Imaging Modalities in Ophthalmic Drug Delivery Studies

Molecular imaging techniques allow for the visualisation, characterisation and quantification of the biological procedures at the cellular and subcellular level [14,15], and their advantage over conventional study methods is that it is possible to take measurements from the same animal at different times without causing harm and with enough spatial and temporal resolution, so as to study different biological procedures in real time. Furthermore, it is possible to conduct a repetitive, uniform and relatively automatic study of the same living subject using identical or alternating biological images at different times. Thus, we can take advantage of the statistical power of longitudinal studies and reduce costs [14,16,17]. Moreover, it complies with the 3Rs principle of animal research proposed by Russel and Burch in 1959: replacement, reduction and refinement [18]. The current molecular imaging modalities include optic imaging-based techniques, positron emission tomography (PET), single-photon emission computed tomography (SPECT) and some magnetic resonance imaging sequences (MRI) [19]. Among this wide variety of tests, there is no perfect modality presenting at the same time high sensitivity, specificity, and time and spatial resolution; however, it is possible to study biological procedures when combining the information provided by each of them. The combination of two or more modalities allows for obtaining images of the same biological procedure in the same animal [20]. The most frequent combinations are those of high sensitivity with anatomic/morphologic techniques. Thus, the combinations commonly used are PET and SPECT, with great sensitivity, along with morphologic techniques like computed tomography (CT) scan and MRI [16]. 

## 3. Positron Emission Tomography

Positron emission tomography (PET) is a relatively new imaging modality mainly used to study brain metabolism, heart function and tumour detection [21]. The PET scanner detects photons emitted by a radiopharmaceutical (molecule of interest which has been previously labelled with a radioisotope), in a way that three-dimensional images are obtained from the concentration of the molecule throughout the body by means of data reconstruction [22,23]. Different radioisotopes can be used to create radiopharmaceuticals. The most commonly used are typically isotopes with short half-lives, such as ^11^C, ^13^N, ^15^O, ^18^F, ^68^Ga, ^82^Rb, or with longer half-lives, such as ^124^I o ^89^Zr. The ^18^F isotope is most often used in clinical practice, since its half-life is long enough to be able to produce commercial radiopharmaceuticals outside the place of study that can then be sent there [24]. 

This modality has not been widely used in the field of clinical ophthalmology to date [25], although its use is promising for the diagnosis of primary ocular tumours and for neurophysiologic studies [26,27,28,29,30]. In recent years, it has emerged in the research field as an essential tool for the study of the pharmacokinetics of different ophthalmic formulations, since it provides images of the drug’s biodistribution and/or its effect on the eye. Below, its application to the study of intraocular and topical ocular routes of administration will be discussed.

The ophthalmic topical route of administration implies that the drug must overcome some physicochemical, metabolic and biological barriers in order to reach the desired action area. Physicochemical barriers include drug properties such as lipophilia, solubility, molecular size and shape and drug loss from the ocular surface due to nasolacrimal drainage. Metabolic barriers include the activity of cytochrome P450 enzymes in ocular tissues, and the biological barriers are the corneal epithelium, the systemic absorption of the conjunctival sac, the blood-vitreous barrier and the blood-retinal barrier [31].

The PET image provides a quantifiable signal on the pharmacokinetic profile of the topically administered radiopharmaceutical so that it is possible to obtain a signal in the eye after the instillation during the initial times, with a subsequent signal detection in the nasolacrimal duct and in the nasal cavity, given by that part of the formulations eliminated from the lacrimal sac towards the nasal cavity, as we can see in Figure 1 [32]. To do this, two conditions must be taken into account: the drug structure must not be significantly modified when being labelled with the radioisotope, and its effect on the organ function and its metabolism can be assessed through molecular image biomarkers that accumulate in the target organ [16]. 

Through the PET technique, the behaviour of different types of polymers [33,34,35,36], such as chitosan labelled with ^124^I or its combined derivative with N-acetylcysteine (NAC-chitosan) was studied [34]. Another topical ophthalmic hydrogel formulation composed by gellan and kappa-carrageenan gum (0.82% weight/volume) labelled with ^18^F radiotracers was also assessed [33] (Figure 2 and Figure 3). In the aforementioned study, it was shown that it is possible to establish differences between the permanence on the ocular surface for different types of ophthalmic formulations by using different polymers as drug delivery vehicles; the technique proposed could be a safe method for this kind of future studies in humans. Some tacrolimus topical formulations for atopic keratoconjunctivitis [32], econazole-α-cyclodextrin formulated in hydrogel for the treatment of fungal keratitis [37], and cysteamine dispersed in different types of hydrogel for the treatment of cystinosis [38] have also been studied. Furthermore, the PET technique allows for the quantitative analysis of the formulations’ pharmacokinetic profile and the calculation of pharmacokinetic parameters, such as half-life and ophthalmic residence time [38].

With this technology, it is also possible to know the biodistribution of compounds at an intraocular level after their intravitreal administration and their pharmacokinetics (Figure 4 and Figure 5) [39]. It is important to bear in mind that in order to reach a sustained therapeutic concentration of drug in the vitreous, the administration frequency must be based on the drug’s half-life. Its elimination in the vitreous humour can be affected by diverse factors, such as molecular weight, its physicochemical properties, its protein or melanin binding, its metabolism, the surgical procedure, the injected volume or ocular inflammation, as well as the membrane transport mechanisms [40]. 

When in vitro pharmacokinetic studies are carried out, there are a series of factors that are not usually kept in mind, such as the absence of convection in the vitreous [41,42,43,44], the protein and melanin binding or the drug’s metabolism and its active transport [10,45]. Through classic in vivo pharmacokinetic studies of intravitreal injections, these aspects could be solved, but as it has been previously mentioned, they are limited due to the invasiveness of the technique [46,47]. Some models that allow the in vitro simulation of intravitreal pharmacokinetics have been proposed, and they keep in mind anatomy and ocular physiological aspects [45,48,49,50]. PET studies with anti-vascular endothelial growth factor (VEGF) drugs, such as bevacizumab and ranibizumab have also been conducted, and they allowed for the determination of the pharmacokinetic properties of these agents inside the vitreous cavity [29].

## 4. SPECT

SPECT (Single photon emission computed tomography) is a molecular imaging technique that provides a three-dimensional spatial distribution of the administered radiopharmaceuticals. The use of a planar technique known as scintigraphy is common (Figure 6) [51]. This technique follows the same rationale as the SPECT technique, but provides a projected image instead of three-dimensional images [52,53]. 

The radioisotopes used in SPECT images include technetium (^99^mTc), indium (^111^In), iodine (^131^I, ^123^I) and gallium (^67^Ga) [10,52]. In the same way as the PET technique, the SPECT image provides useful information at a clinical and preclinical level by simultaneously combining molecular imaging and anatomical information obtained through CT images. This allows for a precise localisation and quantification of the radiolabelled imaging probe, thereby making it possible to observe the pharmacokinetic profile signal on the animal’s anatomy [32,54]. Its applicability in both animal and human studies turns it into a translational research technique [55]. Like the PET technique, the labelling with a radionuclide must preserve the drug structure, and its effect on the organ’s function and metabolism must be assessable through molecular image biomarkers [16].

Gamma scintigraphy was first assessed by Rossomondo et al. in 1972 [56], and was later used with modifications in preclinical studies [57] and in humans [58,59,60,61,62] to assess the lacrimal drainage and the lacrimal gland inflammation [63]. Subsequently, it has been used with the aim of understanding the ocular surface biodistribution of different formulations used as drug delivery vehicles in mice and humans, as it is the case for artificial lacrimal products that contain hydroxypropyl methylcellulose (HPMC), a combination of PVA and gellan gum for antiglaucoma drug release [64,65], PVA [66,67,68] or sodium hyaluronate [69]. Furthermore, with this technique the biodistribution and permanence on the ocular surface of ophthalmic ointments [70], liposomal formulations [71,72], nanoparticles [72] and microemulsions w/o [73] have also been studied. SPECT has been chosen for multiple ocular biopermanence studies of gels and gelation systems, such as carbomer [74] and chitosan-HPMC [75,76]-based gels, thermosensitive poloxamer gels (Pluronic® F-127 and Pluronic® F-68 in combination) [77], in situ ion-sensitive and thermosensitive gelation systems based on a mixture of poloxamer (Pluronic® F-127 and Pluronic® F-68) and chitosan for ofloxacin release [78]. Also, an in situ gelation system activated with ions and based on alginate and HMPC [79], another one activated by ions and pH-based in gellan gum and chitosan [80] and a ion-sensitive gel consisting of gellan gum and kappa-carrageenan [81] were studied. The SPECT/CT technique has made it possible to study the ocular biodistribution of 3-^123^I-iodochloroquine, injected intravenously in rats [82], allowing the non-invasive quantification of drugs in the eye throughout the time and the melanin influence on the ocular distribution and elimination.

SPECT images of the intravitreal kinetics of HSP-70 protein in the retinal pigment epithelium of ex vivo bovine eyes have been obtained, as well as measurements of its cytoprotective effects against oxidative stress, which has appeared as a possible strategy against the degeneration of the retinal pigment epithelium [83].

Both PET and SPECT techniques present an elevated quantification sensitivity, although the values reached through PET (about nM) are higher than those obtained through SPECT. Both techniques provide three-dimensional images that enable the interpretation and quantification of the results in relation to other types of techniques, and even more so when they are combined with CT images, which provide high-resolution anatomical information.

Moreover, as previously mentioned, the labelling of the molecules to be studied can change the properties of some compounds, resulting in alterations in biodistribution. This especially occurs in the case of small molecules. On the contrary, the pharmacokinetics of proteins, large molecules and drug delivery systems are less sensitive to this kind of modifications [10]. In the case of ocular studies, the blood-retinal barrier is an important consideration, since changes in the molecular size and lipophilia can alter its permeability and cause changes in the vitreous clearance. The typical strategies include the use of peptide bonding elements in the labelling of proteins [84], the encapsulation of hydrophilic or lipophilic compounds in liposomes or nanoparticles [85,86], radiometal radiotracers with DOTA complexes (1,4,7,10-tetraazaciclododecane-1,4,7,10-tetraacetic acid) [84]. The studied molecules are considered relatively stable since they can carry out the ligand synthesis in situ and present relatively long physical half-lives (that last from hours to days) [87,88]. It is important to bear in mind that, as previously stated, the labelling of the studied molecules must not significantly affect their structure, which could be translated into changes in its biodistribution and pharmacokinetics.

Another advantage is that the radioactive signal is not affected by changes in the environment, so it provides an excellent depth and a dynamic linear range of the measurements [10]. Regarding PET, the most common and important biological elements (carbon, oxygen, nitrogen and fluorine) have positron-emitting isotopes, which facilitate the labelling of a wide variety of compounds through covalent bonds, leading to great versatility in the development of PET biomarkers in comparison with the ^99^mTc (attached via bulky chelating linker groups) or ^123^I (which forms a biologically unstable bond with carbon) used in the SPECT technique. The molecular structure and chemical properties of the final molecule will be, in many occasions, almost identical to the initial one when it is possible to directly substitute the stable isotope for the radioisotope [16].

The main advantage of SPECT over PET is the possibility to use and detect a variety of radioactive agents based on radioisotopes with different energies, which allows for simultaneously visualising two or more molecular routes. Additionally, SPECT radionuclides have more specific activity in comparison with PET tracers and they are more long-lasting, which makes their application more economical [89]. The use of radiotracers in much lower dosages than the pharmacological ones (in the nano- or microgram range) allows for the safe acquisition of images without the interference of the radiotracer in the studied biological processes. Thus, they present similar pharmacokinetic characteristics but in the range of absence of therapeutic effects, while pharmacological dosages are in the milligram or gram range and are associated with pharmacodynamical effects. In this way, the lowest possible masses of injection compounds to be studied are obtained, guaranteeing the pharmacodynamical inertia of the image-creation process [16].

The injected dose of radiotracers in pharmacokinetics assays depends on the half-life of the radioisotopes, sensitivity of technique and the design of the study (timing, duration, etc.). Thus, in ocular studies with PET and SPECT different activities have been used from 10 MBq (124I, half-life = 4.17 days) [35,90] to 1 MBq (99mTc half-life = 6 h) [34]. Obviously, this issue is especially relevant for the clinical implementation of these procedures due to the fact that a detailed assessment of the radiation protection of the patient must be performed previously. Eye lenses have very high radiosensitivity when compared with other organs and tissues and radiation doses around 2 Gy leads to the development of cataracts. A calculation of the dose delivered to the eye lens in a clinical study using the dose of radiotracers employed in preclinical studies provided values of radiation doses in the range of 10–20 mGy, which are significantly below the dose limits [33]. Molecular imaging has been widely used in clinical ophthalmology but the specific application for ocular pharmacokinetic studies is not still considered as reference for the design of pharmacokinetic clinical trials.

Another limitation of PET and SPECT images is their low spatial resolution (0.5–2 mm), which only allows visualizing the eye without details of tissues such as the retina. This becomes a problem for the delineation of the vitreous area, which implies that the measurements cannot be exclusively restricted to the area of the vitreous [39]. For this reason, it is necessary to combine the computed tomography scan and the MRI for the anatomic localisation of tissues.

## 5. Fluorescence

The optic fluorescence-based image has been used in ophthalmology since the 1980s. The original in vivo fluorometers, were developed as clinical instruments which aimed at determining the lacrimal flow rate, the aqueous renovation, the blood-retinal barriers (in diabetes) and the early formation of cataracts. The fluorometer allows the detection of fluorophore emission signals (e.g., fluorescein) after excitation through the transparent ocular tissues (e.g., cornea, aqueous humour, different depths of the visual axis through time). Fluorophotometry is limited to the evaluation of the movement of fluorescent compounds in the aqueous and vitreous humour. This technique has been used in vivo to control fluorescence levels, mainly in humans and rabbits [91]. 

Nowadays, there are various fluorophores available (such as Alexa Fluor dyes) that are compatible with in vivo fluorophotometers wavelengths. They have improved the brightness, the stability in the presence of light excitation and allow for more robust fluorescence spectrums in changing environments [92]. These features provide a dynamic range with longer linearity (fluorescence emission versus drug concentration) that allowed for the intravitreal pharmacokinetic control of ranibizumab in a 20 times concentration range [93].

This technique has been used for the development of an in situ gelation system Gelrite/alginate-based activated by ions for the ophthalmic administration of matrine, in which the time of permanence on the ocular surface was measured in human volunteers [94]. Some studies correlated the rheology and precorneal retention time in humans, proving that formulations based on gellan gum remained on the ocular surface for 1–3 h (depending on the polymer concentration), while poloxamer and carbomer formulations remained for less than 1 h [95,96,97]. 

As previously mentioned, the ranibizumab intravitreal kinetics have been recently studied using in vivo fluorophotometry (AlexaFluor tracer 488) and ELISA, being obtained pharmacokinetic parameters similar with both approaches. This study proved that it is crucial to use a minimum number of fluorophores for labelling, and select those with brightness, photosensitivity and sensitivity at minimal pH levels. It is possible to measure the effects of drugs and delivery systems on the blood ocular barrier by following the entrance of intravenous FITC-dextran to the eye. This method was demonstrated to be used to study transscleral drug delivery [98,99,100].

The main advantage of this technique lies in its capacity to provide optical images with useful information about the pharmacokinetics of macromolecules and drug delivery systems in the eye [10,93]. One of the potential uses of this technique is the monitoring retention of drug delivery systems on both the ocular surface and vitreous. Currently, there are new in vivo fluorometers available that are more robust than the classic ocular in vivo fluorometer. This would allow potentially expanding the use of this methodology [10].

On the other hand, one of the main disadvantages of in vivo fluorophotometry is quantification accuracy, since the fluorescence signal depends on geometrical factors and fluorescence probe properties. The fluorescence emission depends on the medium (e.g., pH), the concentration (not linear relationship between signal-fluorophore concentration) and light exposure (photo-quenching) [101]. 

Moreover, only a few drugs present intrinsic fluorescence spectrums that would allow for obtaining non-invasive fluorometric images. Drugs that do not have intrinsic fluorescence should be labeled. The stability is usually one of the critical points for the correct development of the trial [10]. 

## 6. Magnetic Resonance Imaging

Magnetic resonance imaging (MRI) is another imaging technique used for the study of ocular structures [102,103], pathologies [104,105] and ophthalmic drug delivery [11,106]. It is a non-invasive technique, like the previous ones, that provides transversal images from the inside of living organisms [107]. It allows for real-time determination of the distribution pattern of the compound of interest in the eye, without disturbing the tissue or the redistribution of said compound [11].

This method is based on nuclear magnetic resonance signals emitted by magnetically labelled nuclei (for example, Gd) [102]. The MRI depends on the magnetic properties of the tissues and its interactions with strong external magnetic fields [16]. It is a very versatile technique that provides morphological images with excellent contrast and spatial resolution (<50mm for preclinical devices and 300mm in experimental clinical devices of ultra-high field) [108]. Thus, it is possible to obtain information about tissue composition, perfusion, oxygenation, tissue elasticity, metabolism, and to detect molecular probes in just one acquisition session without radiation exposure [109]. 

MRI has been used in preclinical studies to determine permanence on the ocular surface of ion-sensitive hydrogels based on gellan gum and kappa carrageenan to be employed as vehicles for ophthalmic drug delivery [110] (Figure 7). Through this study, Ferreiro et al. assessed the time of in vivo corneal contact and the hydrogel residence place after its administration, showing the absence of liquid solution 30 minutes after delivery, whereas in the case of the gellan gum formulation, it was still present three hours after delivery. The main improvement of this method is the absence of a contrast agent. In fact, the large quantity of water retained in the gel increases the intensity of the signal and, therefore, MRI seems to be a good and relevant method for assessing the precorneal retention time of the in situ gelation delivery systems.

In addition to the studies on animals, Bert et al. [111] studied solute diffusion through the anterior route in the human eye by using dynamic magnetic resonance with contrast. The pilocarpine effects on the blood-aqueous barrier after its topical application have been assessed in humans [112].

Diverse studies have demonstrated the usefulness of the MRI for characterising penetration routes, ocular barriers and dynamic procedures [113,114], transscleral penetration routes, mechanisms that allow for an improved drug delivery flow, the location of periocular and intraocular devices (or implants) and its release kinetics in periocular, intrascleral, suprachoroidal and intravitreal administration, as well as the study of the ocular iontophoresis [106,115,116,117,118,119,120,121,122,123,124,125]. With the appropriate MRI contrast agents, it is possible to study the blood-aqueous barrier and the clearance in the anterior chamber, being this an important surmise that contrast agent clearance and water clearance in the aqueous humour are the same. The difference between them may lead to errors in the kinetic assessment of the aqueous humour [11].

MRI can be used in the posterior segment of the eye in order to examine the blood-retinal barrier and the vitreous fluidity in a non-invasive way [126,127,128,129]. The dysfunction in this barrier can be studied through MRI by using a contrast agent such as GdDTPA^2−^, intravenously administrated into the retinal blood vessels and by controlling its dysfunction in the vitreous humour, which allows for quantifying the barrier’s permeability after inducing lesion in the retina [104,129,130]. Similar studies have determined the effects of the endotoxininduced endophthalmitis and the experimental diabetic retinopathy in the blood-retinal barrier function [131,132], as well as ischemic-induced injuries [133] and their hyperpermeability after the sustained elevation of the vascular endothelial growth factor (VEGF) in the vitreous cavity [134]. For this reason, the MRI is a technique with potential for the study of ocular drug delivery systems for the treatment of posterior ocular diseases related to the blood-retinal barrier function.

In the intraocular drug delivery field, the pharmacokinetics of Gadolinium-DTPA after polymeric intraocular implants administration in rabbits have been studied with the MRI technique [115,123]. Furthermore, the biodistribution of magnetic micro- and nanoparticles subsequent to intraocular delivery has also been studied [135]. Nowadays, the dynamic contrast images modality (DCE-MRI) is the most commonly used for preclinical and early clinical assessment of anti-angiogenesis inhibitors like bevacizumab, sorafenib or sutinib [136].

One advantage of the MRI over other molecular imaging techniques is its excellent spatial resolution, but its usefulness in ocular pharmacokinetic studies is limited, as the quantification and sensitivity linear range (approximately 0.1 mM) is rather limited in comparison to radioimaging techniques [10]. MRI has a sensitivity from three to six orders of magnitude lower than the PET technique [137]. Thus, nowadays MRI images are not an attractive option for ocular pharmacokinetics studies. Nevertheless, in association with SPECT and PET, it can be a useful technique in order to improve radiolabelled molecules’ localisation thanks to the anatomical information provided therein.

The majority of the contrast agents used in MRI are hydrophilic filler compounds that can establish specific interactions with the eye, potentially interfering with ocular clearance and pharmacokinetics in permanence time characterisation studies of hydrogels on the ocular surface [138].

One of the advantages that MRI offers over the PET and SPECT radiolabelling techniques is that γ radiations emit relatively low energy levels in comparison to α and β radiations, at the same time as the ^99^mTc exhibits a relatively short half-life (6 h). This is translated into a low risk for health associated with studies with this technique, both for trial subjects and for researchers [139]. Studies on mice have proven the absence of toxicity at the tissue level in epithelium and cornea with the use of Gd-DTPA as a contrast agent [125]. In addition to this, the magnetic resonance imaging projection is not affected by medium opacities such as cataracts [89].

## 7. Other Imaging Techniques

### 7.1. Ultrasonography

Ultrasound imaging uses acoustic transducers that send and receive ultrasound frequency energy to generate reflexion or direct transmission three-dimensional images [140]. Ultrasonography is the most used clinical imaging modality due to its low cost, availability and safety, and it is ideal for obtaining in vivo images in both animals and human patients. The image contrast depends on the image generation algorithm and its use of the backscatter, the sound reduction and the sound speed [141,142]. In ophthalmology, ultrasound probes of 10, 20, 40 and 50 MHz are used. However, the resolution is still the limiting factor. Recent advancements in imaging technology with high-resolution ultrasound devices and the development of specific microbubble contrast agents are promising for the future of molecular imaging [141,143].

In ophthalmology, this technique has been used in diagnostic procedures [144], in order to boost drug administration on the ocular surface (such as tobramycin and dexamethasone) [145,146], protein-loaded nanoparticles [147] and intravitreal administration [148], and to obtain information about juxtascleral administration [122].

### 7.2. Optical Coherence Tomography and Computed Tomography

The recent introduction of new imaging techniques has improved many aspects of the ophthalmic procedures, both therapeutically and diagnostically. This is the case for retina diagnostic techniques: optic coherence tomography (OCT) and OCT with angiography (OCTA) [149]. OCT generates images, interferometrically measuring the amplitude and the delay of the reflected or backscattered light, while OCTA also allows for the visualisation of the macular capillaries and the optical nerve head [150,151]. This technique has been used in comparative studies for artificial tears [152,153]. Its use is based on the measurement of the corneal thickness before and after topical formulation administration at different times. From the data obtained, it was observed that the hyaluronic acid increases the thickness of the lacrimal layer for 30 min in healthy subjects, showing good reproducibility.

CT is an imaging technique that does not require radioactive labelling or contrast, but whose functioning is based on the emission of a large series of two-dimensional X-ray images taken around a single rotation axis [141,154]. This technique has also been used in ocular studies, but as a tool for the anatomic support in the interpretation of images obtained through gammagraphy or PET [155]. Combined with these techniques, it provides an image of the subject’s anatomy at the same time that the image of the radiologically labelled compound provides a quantifiable signal on the pharmacokinetic profile, as we can see in Figure 8 [32]. 

Table 1 summarizes imaging agents used in each technique and their administration, topical or intraocular and the spatial resolution achieved with each technique.

## 8. Conclusions

Molecular imaging techniques make it possible to obtain reliable pharmacokinetic data with the objective of improving the application and permanence of the ophthalmic drugs in its action site, leading to dosage optimisation. The disadvantages of the classical methodologies were the difficulty in obtaining information about topical and intraocular distribution and clearance, associated with multiple factors related to ophthalmic physiology, as well as complexity and invasiveness intrinsic to the sampling techniques. Contrary to these studies, the molecular imaging techniques allow for the real-time visualisation, localisation, characterisation and quantification of the compounds after their administration, all in a reliable, safe and non-invasive way. None of these techniques presents simultaneously high sensitivity and specificity, but it is possible to study biological procedures with the information provided when the techniques are combined. Therefore, molecular imaging techniques are postulated as a resource with great potential for the research and development of new drugs and ophthalmic delivery systems.

## Figures and Tables

**Figure 1 pharmaceutics-11-00237-f001:**
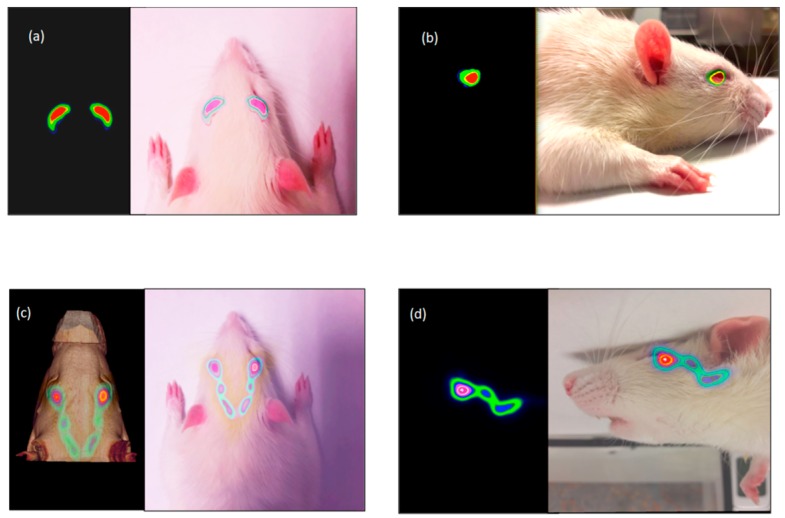
Fusion PET-CT-Real images of the rat’s head. Tacrolimus eye drops remain on the ocular surface after instillation (**a**,**b**) subsequently observed as it is eliminated by the nasolacrimal ducts (**c**,**d**). Axial (**a**) and sagittal (**b**) PET image and fusion PET-real images (10 min). Axial (**c**) fusion PET-CT image and Sagittal (**d**) fusion PET-real images (90 min post-administration). Reproduced with permission from [32], published by Elsevier, 2018.

**Figure 2 pharmaceutics-11-00237-f002:**
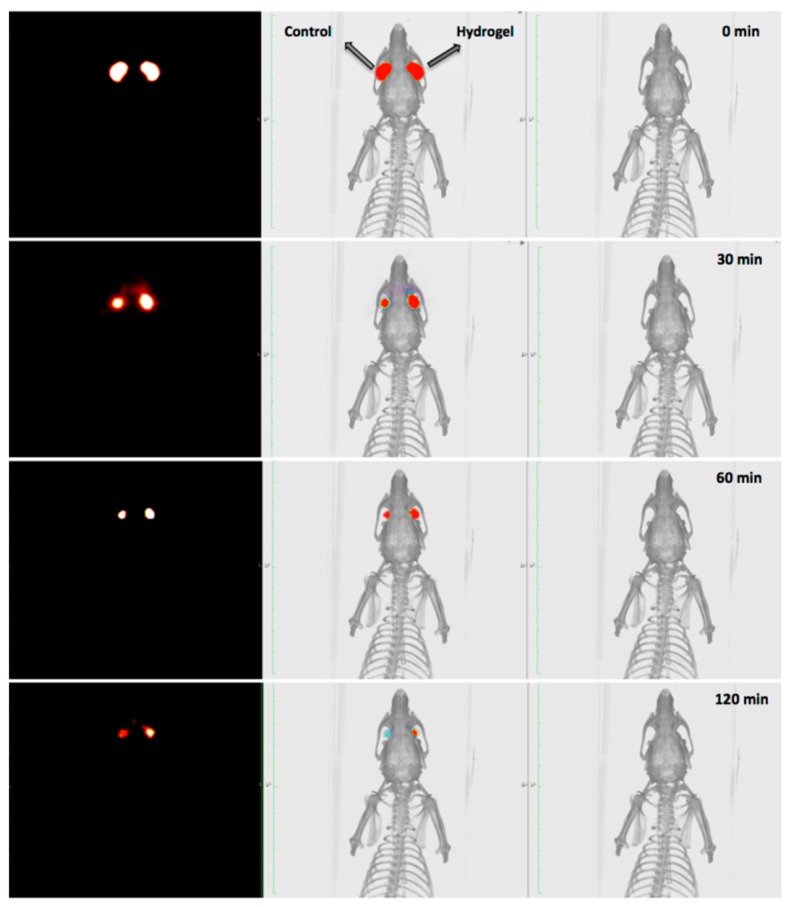
PET (left), CT (right), and PET/CT fused image. The biopermanence evaluation of the hydrogel labeled with 18F-fluorodeoxyglucose (FDG) (right eye) in rats compared to a saline solution containing 18F-FDG as control formulation (left eye). Reproduced from [33], which is licensed under a Creative Commons Attribution-NonCommercial-NoDerivatives 4.0 International License.

**Figure 3 pharmaceutics-11-00237-f003:**
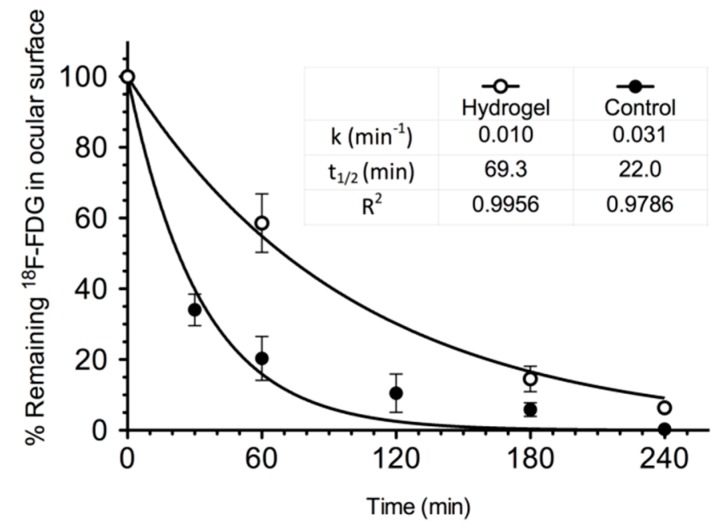
Kinetic profile of topical hydrogel labeled with 18F-FDG (blank circles) compared to a saline solution containing 18F-FDG as control solution (solid circles). Reproduced from [33], which is licensed under a Creative Commons Attribution-NonCommercial-NoDerivatives 4.0 International License.

**Figure 4 pharmaceutics-11-00237-f004:**

Fused image PET/CT showing the signal evolution of intravitreal injections in the rat eyes throughout time. Reproduced from [39], which is licensed under a Creative Commons Attribution-NonCommercial-NoDerivatives 4.0 International License.

**Figure 5 pharmaceutics-11-00237-f005:**
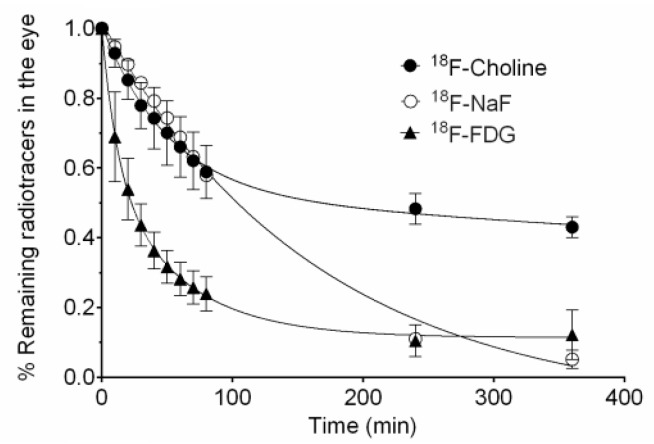
Intravitreal pharmacokinetic profile of ^18^F-FDG, ^18^FNaF, and ^18^F-Choline after intravitreal injection. Reproduced from [39], which is licensed under a Creative Commons Attribution-NonCommercial-NoDerivatives 4.0 International License.

**Figure 6 pharmaceutics-11-00237-f006:**
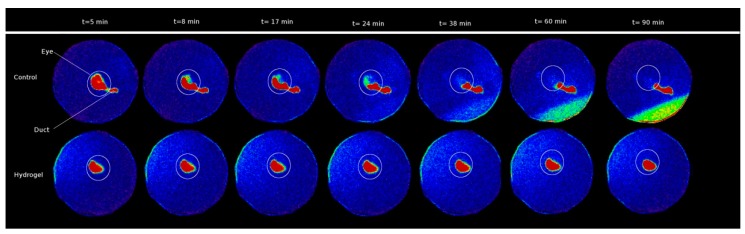
Scintigraphy image of a novel ion-sensitive hydrogel based on gellan gum and kappa-carrageenan. In the upper images (control eye), the signal decreases in the corneal surface, moving towards the nasolacrimal duct, while in the bottom images, signal obtained in the eye with hydrogel remains unchanged. Reproduced with permission from [51], published by Elsevier, 2017.

**Figure 7 pharmaceutics-11-00237-f007:**
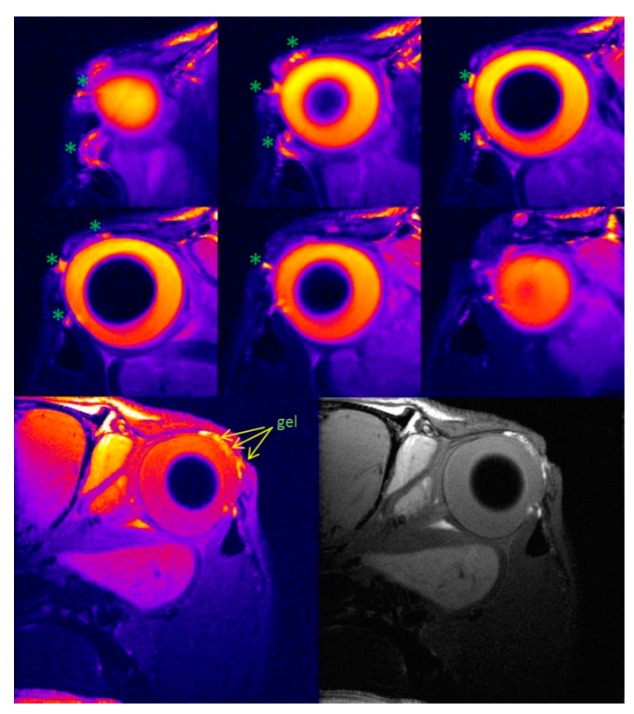
MRI of the rat eye rat 2 h after instillation of ion-sensitive hydrogel indicate the hyperintense signals caused by hydrogel. Reproduced with permission from [110], published by Elsevier, 2015.

**Figure 8 pharmaceutics-11-00237-f008:**
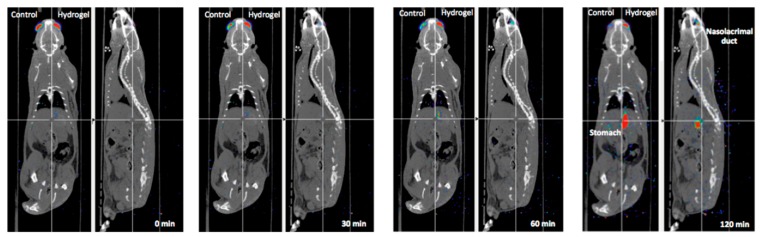
Fused whole-body PET/CT images (coronal and sagittal views). Reproduced from [33], which is licensed under a Creative Commons Attribution-NonCommercial-NoDerivatives 4.0 International License.

**Table 1 pharmaceutics-11-00237-t001:** Administration routes used in each molecular imaging technique, their imaging agents and spatial resolution.

Technique	Administration	Imaging Agents	Spatial Resolution	Reference
**PET**	topical	^18^F, ^124^I	1–2mm	[31,32,33,34,35,36,37,38]
	intraocular	^18^F, ^124^I		[29,39]
**SPECT**	topical	^99m^Tc,^131^I, ^123^I, ^111^In, ^67^Ga	0.5–2mm	[61,62,63,64,65,66,67,68,69,70,71,72,73,74,75,76,77,78,79,80]
	intraocular	^123^I		[82]
**Fluorescence**	topical	Fluorescein, AlexaFluor 488	1–10mm	[94,95,96,97]
	intraocular	Fluorescein, AlexaFluor 488		[98,99,100]
**Magnetic resonance imaging**	topical	Gadolinium	25–100um	[109,110]
	intraocular	Gadolinium		[114,122,134]
**Ultrasonography**	topical	Microbubbles	30–100um	[144,145,146,147]
**Optical coherence tomography**	topical	None	10–20um	[151,152]

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
