# Peer review of "Ocular Biodistribution Studies Using Molecular Imaging"

_pharmaceutics, 2019, doi:10.3390/pharmaceutics11050237_

Round 1

Reviewer 1 Report

The author reviewed molecular imaging technologies in the application of ocular bio-distribution studies, including PET, SPECT, optical imaging and so on. For the review paper, I have some questions list as below:

If possible, could you show the normal dosage range used in the PET, SPECT, MRI etc. 

How about the resolution in the modalities?

How to correlate probe amount with radioisotope declining (take account of half-life f radioisotope itself)

In figure2, please indicate the probes used. What is the control, to make readers understand easily

From line 325-330, I felt confused that why fluorescence can not be used to monitor pharmacokinetic? People used  fluorescence a lot to do that for pd/pk investigation. Also, please think it over about the 2.4.3, the advantages and disadvantages description seems not accurate for me.

People used small molecules and macromolecules a lot in the application of molecular imaging. If possible, please give some descriptions that what is the advantages and disadvantages between them (ocular studies)

Author Response

Reviewer #1:

Thank you very much for your valuable comments. We greatly appreciate all your corrections, which have clearly improved our manuscript.

If possible, could you show the normal dosage range used in the PET, SPECT, MRI etc.

We have added the following text in the manuscript:

The injected dose of radiatracers in pharmacokinetics assays depends on the half-life of the radioisotopes, sensitivity of technique and the design of the study (timing, duration…). Thus, in ocular studies with PET and SPECT different activities have been used from 10 MBq (124I, half-life= 4.17 days) [34,87] to 1 MBq (99mTc half-life = 6 hours) [33]. Obviously, this issue is especially relevant for the clinical implementation of these procedures due to that a detailed assessment of the radiation protection of the patient must be performed previously. The eye lens have very high radiosensitivity when compared with other organs and tissues and radiation doses around 2 Gy leads to the development of cataracts. A calculation of the dose delivered to the eye lens in a clinical study using the dose of radiotracers employed in preclinical studies provided values of radiation doses in the range of 10-20 mGy, which are significantly below the dose limits [32]. Molecular imaging has been widely used in clinical ophthalmology but the specific application for ocular pharmacokinetic studies is not still considered as reference for the design of pharmacokinetic clinical trials.

How about the resolution in the modalities?

The spatial resolution of the different modalities is as follows: PET and SPECT (0.5-2mm), MRI (25-100 um), fluorescence (1-10mm), CT (50-100 um), Ultrasonography (30-100um) and OCT 10-20um. This information was incorporated into Table 1.

How to correlate probe amount with radioisotope declining (take account of half-life f radioisotope itself)

For these studies, different regions of interest (ROIs) are manually drawn containing the signal on each eye. The ROIs are then replicated on the different temporal image frames to obtain the decrease curve of the radioisotope over time, conveniently corrected for radioactive decay:

N(t1/2) = N0. e -λt/t1/2

t1/2: Half-life

t : Time elapsed

N: Power of dose to be calculated (decayed)

N0: is the initial number (when t = 0) of nuclei in the sample. Initial dose power (calibrated)

λ: disintegration constant (λ = -0.693).

In figure 2, please indicate the probes used. What is the control, to make readers understand easily.

New text has been added in Figure 2. “The biopermanence evaluation of the hydrogel labeled with 18F-FDG (right eye) in rats compared to a saline solution containing 18F-FDG as control formulation (left eye)”

From line 325-330, I felt confused that why fluorescence can not be used to monitor pharmacokinetic? People used fluorescence a lot to do that for pd/pk investigation. Also, please think it over about the 2.4.3, the advantages and disadvantages description seems not accurate for me.

Thank you for such wise appreciation. We have modified this section because our intention is not to express that it was not useful for pharmacokinetic studies.

People used small molecules and macromolecules a lot in the application of molecular imaging. If possible, please give some descriptions that what is the advantages and disadvantages between them (ocular studies)

This point has been detailed in lines 260 – 299.

Reviewer 2 Report

The manuscript presents the advancements on the ocular distribution studies based on molecular imaging tools. The article was concise and well written, focusing on imaging modalities for demonstrating the ocular distribution. In my opinion, it could be publishable on pharmaceutics after an ample amount of revision. Here are some comments below.

In the abstract, better add some details based on previously used techniques and advanced techniques for better insight. It was also confusing that recent developments were discussed first and their respective methodologies later, better exchange the writing, would make it logical and insightful to the reader.

Not necessarily required to have subsections naming as advantages and disadvantages, as well as topical and intraocular routes, better discuss the topic under a single heading.

Add some discussions and tabulated form on the scope for clinical translation of imageable agents or composites for ocular distribution along with the phase of the trial.

Any schematic showing the outline would be insightful.

Add some recent literature (in 2019) relevant to the field

Author Response

Reviewer #2:

Thank you very much for you contribution in the manuscript improve. We greatly appreciate the constructive review.

In the abstract, better add some details based on previously used techniques and advanced techniques for better insight. It was also confusing that recent developments were discussed first and their respective methodologies later, better exchange the writing, would make it logical and insightful to the reader.

Thank you for the appreciation. The proposed changes have been introduced in the abstract.

Not necessarily required to have subsections naming as advantages and disadvantages, as well as topical and intraocular routes, better discuss the topic under a single heading.

Those subsections of the manuscript have been removed.

Add some discussions and tabulated form on the scope for clinical translation of imageable agents or composites for ocular distribution along with the phase of the trial.

New text was added in the manuscript. Thank you for the commentary, it´s very interesting.

The injected dose of radiatracers in pharmacokinetics assays depends on the half-life of the radioisotopes, sensitivity of technique and the design of the study (timing, duration…). Thus, in ocular studies with PET and SPECT different activities have been used from 10 MBq (124I, half-life= 4.17 days) [34,87] to 1 MBq (99mTc half-life = 6 hours) [33]. Obviously, this issue is especially relevant for the clinical implementation of these procedures due to that a detailed assessment of the radiation protection of the patient must be performed previously. The eye lens have very high radiosensitivity when compared with other organs and tissues and radiation doses around 2 Gy leads to the development of cataracts. A calculation of the dose delivered to the eye lens in a clinical study using the dose of radiotracers employed in preclinical studies provided values of radiation doses in the range of 10-20 mGy, which are significantly below the dose limits [32]. Molecular imaging has been widely used in clinical ophthalmology but the specific application for ocular pharmacokinetic studies is not still considered as reference for the design of pharmacokinetic clinical trials.

Any schematic showing the outline would be insightful.

The elimination of the subsections clarifies the review. In this way, it is only divided into 8 final sections.

Add some recent literature (in 2019) relevant to the field

Three new references (2019) have been added in the manuscript:

-        [25]: Pulagam KR; Gómez-Vallejo V; Llop J; Rejc L Radiochemistry; a useful tool in the ophthalmic drug discovery. Current Medicinal Chemistry 2019.

-        [62]: Kim, D.J.; Baek, S.; Chang, M. Usefulness of the dacryoscintigraphy in patients with nasolacrimal duct obstruction prior to endoscopic dacryocystorhinostomy. Graefe’s Archive for Clinical and Experimental Ophthalmology 2019.

-        [63]: Jiang, C.; Li, X.; Zhao, M.; Dend, H.; Huang, J.; Liu, D.; Xu, X. Efficacy of 99mTc-DTPA orbital SPECT/CT on the evaluation of lacrimal gland inflammation in patients with thyroid associated ophthalmopathy. Zhong nan da xue xue bao. Yi xue ban = Journal of Central South University. Medical sciences 2019, 44, 322–328.

Reviewer 3 Report

The review submitted from Castro-Balado and coworkers presents a well-structured, clear and comprehensive overview on the imaging modalities available to investigate biodistribution upon ocular administration.

The contribution itself is thoroughly compiled and provides the reader with valuable reports on the different imaging modalities. Particularly enjoyable is the section "advantages and disadvantages" for each technique. However, too much attention is devoted to the imaging modality itself and too few to the PK and biodistribution, ultimately core topic of the review, at least looking at the title and at the abstract. Complementing the present imaging figures with PK profiles from the same technique would represent an appreciated added value and would help in fulfilling the expectations of the reader. Details from the general introduction dedicated to each single technique could be removed to lighten the text.

Author Response

Reviewer #3:

Thank you very much. We greatly appreciate the constructive review.

The contribution itself is thoroughly compiled and provides the reader with valuable reports on the different imaging modalities. Particularly enjoyable is the section "advantages and disadvantages" for each technique. However, too much attention is devoted to the imaging modality itself and too few to the PK and biodistribution, ultimately core topic of the review, at least looking at the title and at the abstract. Complementing the present imaging figures with PK profiles from the same technique would represent an appreciated added value and would help in fulfilling the expectations of the reader.

New figures with PK profiles have been added.

Details from the general introduction dedicated to each single technique could be removed to lighten the text.

According to the instructions of the reviewer, the dispensable parts of the introduction dedicated to each single technique have been eliminated.

We appreciate your time. We are convinced that your comments have improved the current revision.

Kind regards,

The authors
